# Training Neural Networks with Fixed Sparse Masks

**Yi-Lin Sung**[*]
UNC Chapel Hill
ylsung@cs.unc.edu

**Varun Nair**[*]
Duke University
vn40@duke.edu

**Colin Raffel**
UNC Chapel Hill
craffel@gmail.com

## Abstract

During typical gradient-based training of deep neural networks, all of the model's parameters are updated at each iteration. Recent work has shown that it is possible to update only a small subset of the model's parameters during training, which can alleviate storage and communication requirements. In this paper, we show that it is possible to induce a *fixed* sparse mask on the model's parameters that selects a subset to update over many iterations. Our method constructs the mask out of the $k$ parameters with the largest Fisher information as a simple approximation as to which parameters are most important for the task at hand. In experiments on parameter-efficient transfer learning and distributed training, we show that our approach matches or exceeds the performance of other methods for training with sparse updates while being more efficient in terms of memory usage and communication costs. We release our code publicly to promote further applications of our approach.[2]

## 1   Introduction

Stochastic gradient descent (SGD) is a vital component of the modern pipeline for training deep neural networks. Along with the back-propagation algorithm, gradient descent allows for the efficient minimization of a loss function by gradually updating a model's parameters. SGD minimizes the loss over a small random subset of the dataset at each training iteration, which allows training over large datasets. In practice, minimizing a large neural network's training loss using SGD often results in models that generalize well to new data [3, 17, 27], making SGD an invaluable tool.

While effective, standard SGD requires that all model parameters are updated at every iteration of training. As a result, communicating changes to the model requires communicating the updated value of every parameter. Since modern neural networks often have millions or billions of parameters [9, 7, 40], this communication can become excessively expensive. A concrete example of the negative impacts of these costs arises in the setting of transfer learning. In transfer learning, a model's parameters are initialized from an existing pre-trained model before being fine-tuned (i.e. trained) on a task of interest. Pre-trained models can be fine-tuned a huge number of times – for example, the Hugging Face model repository[3] has thousands of fine-tuned variants of the BERT model [12]. Each of these fine-tuned variants requires a unique copy of the model's parameters, each of which takes up around 500MB of disk space. Relatedly, in distributed training [11] and federated learning [35], workers compute updates for a centralized model in parallel on different subsets of data. After a certain number of updates, the workers each communicate the newly-computed parameter values back to the centralized model. The communication step can cause a significant amount of overhead (particularly when the model is large) since the workers must communicate the updated values of all parameters when using standard SGD.

---

[*]Equal contribution.

[2]Code for our work can be found at `https://github.com/varunnair18/FISH`.
[3]`https://huggingface.co/models`

35th Conference on Neural Information Processing Systems (NeurIPS 2021).

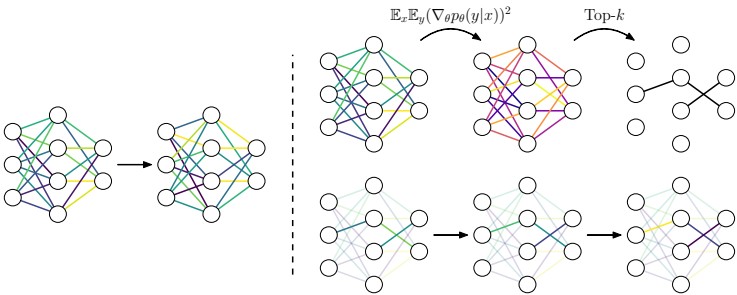

Figure 1: Diagram comparing our proposed method to standard SGD. In traditional gradient-based training (left), all of a model's parameters are updated at every iteration. We propose FISH Mask, a method for precomputing a sparse subset of parameters to update over many subsequent training iterations. To construct the FISH Mask, we find the $k$ parameters with the largest Fisher information (right, top). Then, we train the model with traditional gradient descent, but only update those parameters chosen by the mask (right, bottom).

These issues could be mitigated if it was possible to only update a few parameters during training while still maintaining performance close to that of training all parameters. This has led to various work on parameter-efficient training of neural networks. For example, Adapters [19, 41, 5] introduce additional parameters into a pre-trained model in the form of small task-specific modules that are fine-tuned while the rest of the model's parameters are kept fixed. Diff Pruning [16] and BitFit [6] demonstrate that it is possible to fine-tune a model while only updating a small subset of the existing parameters. In distributed and federated learning settings, Aji and Heafield [2] and Konečný et al. [23] have shown that it is possible for each worker to only update a sparse subset of a model's parameters, thereby reducing communication costs.

Existing methods for training with sparse updates typically work in one of three ways: they either add parameters to the model (Adapters), choose a hand-defined and heuristically-motivated subset of parameters (BitFit), or allow the subset of parameters to change over the course of training (Diff Pruning and methods in distributed and federated training). In this paper, we argue for *pre-computing* a sparse subset of existing parameters to update and keeping the subset *fixed* over many iterations of training. This approach yields various benefits: First, by updating a subset of existing parameters instead of adding parameters (as is done in Adapters), we avoid any increase in the total size of the model. Second, by avoiding hand-defining the mask, we can ensure that our procedure is model-agnostic. Third, by pre-computing a mask, we avoid the computational and memory overhead that are apparent when updating the mask over the course of training. It also allows workers in the distributed training setup to operate on strictly complementary subsets of parameters. Finally, keeping the mask fixed over many iterations, we can ensure that only a specific fixed number of parameters are updated. We are not aware of any existing techniques that satisfy these desiderata.

Motivated by these benefits, we introduce a new method for pre-computing fixed sparse masks. Our approach first estimates the importance of each parameter using an empirical approximation of its Fisher information. Then, we construct the mask by choosing the $k$ parameters with the largest Fisher information. The resulting mask, which we deem a "FISH (**F**isher-**I**nduced **S**parse unc**H**anging) mask", can be re-used for many subsequent training iterations. We demonstrate the effectiveness of using a FISH Mask in a wide variety of settings, including parameter-efficient transfer learning, distributed training with long delays across workers, and reducing checkpoint size. Broadly speaking, FISH Mask training can dramatically reduce storage and communication requirements, while sacrificing minimal performance compared to standard gradient descent and outperforming relevant prior methods for training with sparse updates.

## 2 The Fisher-Induced Sparse uncHanging (FISH) Mask

The main contribution of this paper is a method for pre-computing a sparse subset of a model's parameters to update over many subsequent training iterations. To construct such a subset, we use an approximation of each parameter's Fisher information as a signal of how important the parameter is for a given task. We refer to the resulting mask (i.e. binary array indicating which parameters are

included in the subset) as a FISH (**F**isher-**I**nduced **S**parse unc**H**anging) mask. In this section, we provide the necessary background and detail the steps necessary for computing a FISH Mask. This process is diagrammed in fig. 1.

## 2.1 Fisher Information

Our goal is to select the subset of parameters that are (in some sense) the most important to update. One way to measure a parameter's importance is to consider how much changing the parameter will change the model's output. We denote $p_\theta(y|x)$ as the output distribution over $y$ produced by a model with parameter vector $\theta \in \mathbb{R}^{|\theta|}$ given input $x$. One way to measure how much a change in parameters would change a model's prediction would be to compute $D_{KL}(p_\theta(y|x) \ || \ p_{\theta+\delta}(y|x))$, where $\delta \in \mathbb{R}^{|\theta|}$ is a small perturbation. It can be shown [34, 37] that as $\delta \to 0$, to second order,

$$\mathbb{E}_x D_{KL}(p_\theta(y|x) \ || \ p_{\theta+\delta}(y|x)) = \delta^T F_\theta \delta + O(\delta^3) \tag{1}$$

where $F_\theta \in \mathbb{R}^{|\theta| \times |\theta|}$ is the Fisher information matrix [14, 4], defined as

$$F_\theta = \mathbb{E}_{x \sim p(x)} \left[ \mathbb{E}_{y \sim p_\theta(y|x)} \nabla_\theta \log p_\theta(y|x) \nabla_\theta \log p_\theta(y|x)^T \right] \tag{2}$$

Given this relation, it can be seen that the Fisher information matrix is closely connected to how much each parameter affects the model's predictions. Indeed, this has led the Fisher information matrix to be widely used in modern machine learning, e.g. as a measure of parameter importance [21], as a preconditioner in gradient descent [4, 37, 34], as a way to measure the amount of "information" in each parameter of a neural network [1], or as a way to decide which parameters to prune when performing model compression [43, 10, 47].

When applied to large neural networks, the $|\theta| \times |\theta|$ size of the Fisher information matrix makes it intractable to compute. Prior work therefore frequently approximates $F_\theta$ as a diagonal matrix, or equivalently, as a vector in $\mathbb{R}^{|\theta|}$. Separately, when training machine learning models we seldom have the ability to draw samples $x \sim p(x)$; instead, we are given a finite training set of such samples. Furthermore, it is rarely necessary to compute the expectation over $x$ in eq. (2) over the full training set; instead, it can often be well-approximated over $N$ samples $x_1, \ldots, x_N$. These constraints result in the following common approximation:

$$\hat{F}_\theta = \frac{1}{N} \sum_{i=1}^{N} \mathbb{E}_{y \sim p_\theta(y|x_i)} (\nabla_\theta \log p_\theta(y|x_i))^2 \tag{3}$$

where $\hat{F}_\theta \in \mathbb{R}^{|\theta|}$. This approximation also has an intuitive interpretation: A given entry in $\hat{F}_\theta$ relates to the average of the square gradient of the model's output with respect to a given parameter. If a given parameter heavily affects the model's output, then its corresponding entry in $\hat{F}_\theta$ will be large, so we can reasonably treat $\hat{F}_\theta$ as an approximation of the importance of each parameter.

Note that both eq. (2) and eq. (3) include an expectation over $y \sim p_\theta(y|x)$. When the number of classes is small, this expectation can be computed exactly. For tasks with many possible classes, it is common to approximate the expectation with a few samples from $p_\theta(y|x)$. In supervised learning settings, we have access to the ground-truth label $y_i$ for each sample $x_i$ in our training set. This leads to the possibility of replacing $\mathbb{E}_{y \sim p_\theta(y|x_i)} (\nabla_\theta \log p_\theta(y|x_i))^2$ in eq. (3) with $(\nabla_\theta \log p_\theta(y_i|x_i))^2$. Performing this approximation is referred to as the "empirical Fisher". It has been shown that using the empirical Fisher can lead to degenerate behavior when used as a preconditioner in an optimizer [34, 26]. Since our use of the Fisher information is largely based on a heuristically-motivated notion of parameter importance, we experimented with both the empirical and standard (eq. (3)) Fisher approximations and found that they produced similar performance. Furthermore, the Empirical Fisher is faster to compute than the standard Fisher as long as more than one sample is used to approximate the expectation $\mathbb{E}_{y \sim p_\theta(y|x_i)}$. We discuss this further in section 4.4.1.

## 2.2 Computing Fixed Sparse Masks

Recall that our goal is to select a subset of parameters (or, equivalently, a sparse mask over parameters) to update over many iterations of training while keeping the remainder of the parameters fixed. Having established the Fisher information as a useful tool for estimating the importance of a given parameter, we therefore first compute the approximate Fisher information (as described in the previous section)

for all of a model's parameters. Then, to construct the FISH Mask, we simply choose the $k$ parameters with the largest Fisher information, where $k$ is set according to the desired mask sparsity level.[4] Specifically, a FISH Mask comprises the parameters $\{\theta_i \mid \hat{F}_{\theta_i} \geq \mathtt{sort}(\hat{F}_\theta)_k\}$. Computing the FISH Mask is cheap because $\hat{F}_\theta$ can be computed efficiently using backpropagation, and (as we will show in section 4.4.2) we can obtain a reliable mask for relatively small values of $N$. Further, the fact that we re-use the mask for many iterations prevents us from having to compute $\hat{F}_\theta$ frequently. As we will show in section 4, we find that this simple procedure is sufficient to produce a mask that can be reused for many iterations (over 100,000 iterations in some cases) in a wide variety of settings without sacrificing substantial performance compared to standard gradient-based training.

Note that in some applications of transfer learning, a new linear classifier layer must be added to the model to make it applicable to the downstream task. Since the FISH Mask depends on $p_\theta(y|x)$ and is computed before training begins, this means that we must compute the FISH Mask using the randomly-initialized classifier before any training has begun. We find that computing the Fisher information through the randomly-initialized classifier layer still provides a good signal of parameter importance. When applying FISH Mask in transfer learning settings where a new classifier layer is added, we always include the parameters of the classifier in the mask.

## 3 Related Work

Our approach bears similarity and takes inspiration from existing approaches for parameter-efficient transfer learning and distributed training of machine learning models. In this section, we outline related methods, some of which we will compare to directly in section 4. We also briefly describe how our work is related to and differs from work in network pruning.

### 3.1 Parameter-Efficient Transfer Learning

Transfer learning [36], where a model is initialized from a pre-trained checkpoint before being fine-tuned on a related downstream task, can dramatically improve performance and speed up convergence on the downstream task [12, 8, 40]. Standard practice is to update all of the model's parameters during fine-tuning, though in some cases reasonable performance can be attained by only fine-tuning the output layer of the model [20, 8, 38]. Training only the output layer has the benefit that adapting a given pre-trained model to a downstream task only requires adding a relatively small number of new parameters, but typically results in worse performance compared to training the full model [38, 24].

Various methods have been proposed that endeavor to match the performance of fine-tuning the full model while only updating or adding a small amount of parameters. Adapters [19, 41, 5] are small subnetworks that are added between a pre-trained neural networks layers. Various works [19, 33, 32] have shown that, when appropriately designed, updating only the parameters in the adapters and the output layer can approach the performance of fine-tuning all parameters. For example, Houlsby et al. [19] add on average 3.6% more parameters to adapt a pre-trained BERT model [12] to tasks in the GLUE benchmark [48]. Concurrent work by Mahabadi et al. [33] improves the efficiency of Adapters by generating the weights of task-specific adapters via a hypernetwork. A second concurrent approach by Mahabadi et al. [32] introduces COMPACTER, which utilizes matrix decomposition and low-rank parameterization for the adapters' weights. COMPACTER is shown to achieve the same performance as standard fine-tuning of the T5 models [40] while only adding 0.047% as many parameters as the original model. Finally, very recent work has shown that it is possible to train language models to perform a task by only optimizing the parameters of a "prompt" that is injected into the input of the model's layers [29, 28]. This can yield extremely parameter-efficient results (as low as 0.01% task-specific parameters [28]) but this class of methods is only applicable to next-step-prediction language models. The main drawback of all Adapter-style methods is that they increase the parameter count and computational cost of the model. This makes them inapplicable to the distributed training and efficient checkpointing settings consider in this paper. We therefore only compare directly to other methods that do not add any parameters.

More closely related to our approach are methods for choosing a small subset of the model's existing parameters to update. In an extreme case, Zhao et al. [50] find a sparse mask to multiply against

---

[4]To avoid confusion, we use "mask sparsity" rather than "sparsity", as in some works of literature (e.g. network pruning) the latter term has an opposite meaning that denotes the percentage of weights being zero.

pre-trained parameters (which are not otherwise updated). That is, instead of fine-tuning the models, a binary mask is learned that marks which parameters should be zeroed out. The resulting performance degrades heavily when the mask is made very sparse, suggesting that it is likely beneficial to update parameters. More recently, Guo et al. [16] propose "Diff Pruning", where a sparse binary mask is found over the course of training that denotes which parameters should be updated or fixed at the value from the pre-trained model. Mask sparsity in the binary mask is enforced through a smooth approximation of the $L_0$ norm introduced by Louizos et al. [31]. Guo et al. [16] also show improved performance by imposing a structure on the mask according to which parameters correspond to a particular weight matrix or bias vector. Ultimately, Diff Pruning is shown to both be more parameter-efficient and outperform Adapters when applied to fine-tuning BERT on the GLUE benchmark. However, using Diff Pruning requires significantly more memory during training in order to store and update the mask. Another recent result by Ben-Zaken et al. [6] demonstrated that simply updating the bias parameters in BERT can attain competitive performance with Diff Pruning. While this provides a simple and strong baseline, it is not universally applicable – for example, the pre-trained T5 model [40] does not have any bias vectors. In section 4, we show that using a FISH Mask outperforms all of these approaches in parameter-efficient fine-tuning of BERT on GLUE.

## 3.2 Distributed Training

As models and datasets grow, it becomes inefficient or impossible to train a model on a single machine. This has motivated the need for distributed training strategies where computation for training a model is shared across many machines (called workers) [11]. A major consideration in distributed training are communication costs, since workers need to regularly communicate parameter updates with one another. To minimize communication costs, workers can compute multiple updates on their copy of the model before communicating their changes, but this gives rise to the "stale gradient" problem where workers are operating on an out-of-date copy of the model. The standard and straightforward approach to dealing with stale gradients it to simply apply updates in the "wrong" order, which can be effective in practice [11, 42]. An orthogonal approach to reducing communication costs is to have workers only update a small subset of the model's parameters [2, 13, 44, 45]. For example, Aji and Heafield [2] simply have workers communicate only those updates corresponding to the gradients with top-$k$ largest magnitude at each step. This bears a similar motivation to FISH Mask, but results in a "mask" that changes at every iteration and therefore requires workers communicate after each update. In contrast, pre-computing a FISH Mask allows workers to perform multiple iterations before communicating their updates, thereby further reducing communication costs.

An extreme variant of distributed training is federated learning [22, 35]. In federated learning, asynchronous workers perform many updates on private data before communicating the changes back to a centralized model. The training involves one server and multiple clients, and the server model's gradient is the combination of the workers' gradients. As with any form of asynchronous training, communication costs and stale gradients are significant issues. McMahan et al. [35] demonstrated that averaging the updates computed by individual workers is an effective approach to dealing with stale gradients and Konečnỳ et al. [23] investigated techniques for significantly reducing communication costs. Our method is complementary to the techniques for reducing communication proposed by Konečnỳ et al. [23].

## 3.3 Network Pruning

Past work in network pruning has also explored techniques for sparsifying neural networks (i.e. zeroing out many parameters for compression purposes) while sacrificing minimal performance. Most relevant to our work, Theis et al. [46] propose utilizing the Fisher to prune and decrease the overall number of parameters for gaze prediction, and Liu et al. [30] also use the Fisher to discover groups of parameters to prune from common backbone architectures. Critically, these works differ from our work in that we do not train neural networks with sparse weights. Instead, we focus on using the Fisher to inform the selection of a fixed sparse subset of weights in a non-sparse network to update over the course of training.

# 4 Experiments

We evaluate the efficacy of the FISH Mask in three settings: parameter-efficient transfer learning, distributed training, and training with efficient checkpointing. For parameter-efficient transfer learning, we demonstrate that our approach matches the performance of standard gradient-based training on the GLUE benchmark [48] while updating only 0.5% of the model's parameters per task. For distributed training, we evaluate FISH Mask training for both transfer learning on GLUE and training from scratch on CIFAR-10 [25]. In both settings, we find that we can dramatically reduce communication without sacrificing significant performance, though from-scratch training on CIFAR-10 requires a higher mask sparsity level than fine-tuning on GLUE. Finally, we demonstrate a novel application of training with sparse updates: Minimizing the size of checkpoints over training. We show that using a FISH Mask while training on CIFAR-10 with a mask sparsity level of 10% can shrink checkpoint size on disk by a factor of 5 while sacrificing only a small amount of accuracy. Throughout our experiments, we report results with varying mask sparsity to get a sense of the savings induced by the FISH Mask. For ease of comparison, we report mask sparsity in terms of the total percentage of parameters that are updated. This percentage can be converted to a value of $k$ used for the top-$k$ operation when constructing the mask simply by multiplying it against the total number of parameters in the model.

We also include ablation studies to measure the impact of the number of samples used to estimate the Fisher information as well as the choice of true or empirical Fisher. All experiments for GLUE are run with the BERT$_{\text{LARGE}}$ variant of BERT, which contains 16 attention heads, 24 layers, and 330 million parameters in total [12], and most experiments are run on a RTX 3090 GPU. For experiments on CIFAR-10, we use a ResNet-34 [18] with various optimizations for fast convergence.[5] We report the average performance across 5 seeds for all experiments.

## 4.1 Parameter-Efficient Transfer Learning

In parameter-efficient transfer learning, the goal is to fine-tune a pre-trained model while updating as few parameters as possible. We focus on fine-tuning BERT$_{\text{LARGE}}$ on the GLUE benchmark [48], which is the primary setting used for evaluation in prior work. For all experiments, we fine-tune for 7 epochs and perform a hyper-parameter search across learning rate $\in \{1 \times 10^{-4}, 5 \times 10^{-5}, 1 \times 10^{-5}\}$ and batch size $\in \{8, 16\}$ for each GLUE task. We find the learning rate of $5 \times 10^{-5}$ and batch size of 16 to be effective for most tasks, with the exception of batch size $= 8$ used for RTE. Additional hyper-parameters, such as choice of optimizer, sequence length, and others, follow from the default configuration for BERT$_{\text{LARGE}}$ presented in the Hugging Face library [49]. Test set results are reported by submitting to the GLUE benchmark using the final model checkpoint following a hyper-parameter search on validation results, unless otherwise noted.

**Baselines** We compare GLUE task performance of the FISH Mask to several other baselines and methods focused on parameter-efficient transfer learning. In **Dense Fine-tuning**, we fine-tune all parameters of a pre-trained model, as is typical in standard transfer learning. In the **Random Mask** baseline, we randomly select and fix $k$ parameters to update at the start of training. To compare to prior work, we reproduce **Bit-Fit** [6], in which only the bias parameters of the BERT model are updated across training. Our reproduction follows the original paper and performs a hyper-parameter search with learning rates in the $[1 \times 10^{-3}, 1 \times 10^{-4}]$ range. We also reproduce results from **Diff Pruning** [16], which updates the sparse mask over the course of training. Our reproduction of Diff Pruning at 0.5% mask sparsity follows the paper's code-base[6] and training settings, and reports the GLUE test set results using the best validation checkpoint. Due to restrictions on the number of permissible submissions to the GLUE test server, we are only able to report results with a mask sparsity of 0.5% for those methods where we can control the mask sparsity level. We therefore include additional validation set results for varying mask sparsity levels when using a FISH Mask.

**Results** Our results on parameter-efficient transfer learning with the FISH Mask can be seen in table 1. FISH Mask training results in effectively the same performance (82.6%) as standard "dense" fine-tuning (82.5%), despite updating just 0.5% of BERT$_{\text{LARGE}}$ parameters. The Random Mask baseline achieves a significantly lower average GLUE score, which demonstrates the value of using

---

[5]Our implementation is based on `https://github.com/davidcpage/cifar10-fast`
[6]`https://github.com/dguo98/DiffPruning`

| Method | Updated Params/Task | QNLI | SST-2 | MNLI$_m$ | MNLI$_{mm}$ | CoLA | MRPC | STS-B | RTE | QQP | AVG |
|---|---|---|---|---|---|---|---|---|---|---|---|
| Dense Fine-tuning | 100% | 93.4 | 94.9 | 87.0 | 86.1 | 61.0 | 86.6 | 86.5 | 70.9 | 80.5 | 82.5 |
| Random Mask | 0.50% | 89.8 | 93.4 | 83.7 | 84.0 | 43.2 | 77.8 | 87.7 | 61.3 | 77.2 | 76.8 |
| Bit-Fit [6] | 0.08% | 90.4 | 94.5 | 85.0 | 84.8 | 60.3 | 86.3 | 85.0 | 69.6 | 78.5 | 81.2 |
| Diff Pruning [16] | 0.50% | 91.9 | 93.8 | 86.0 | 85.5 | 61.0 | 86.2 | 85.6 | 67.5 | 80.1 | 81.5 |
| FISH Mask | 0.08% | 93.3 | 94.0 | 85.3 | 84.9 | 56.4 | 86.2 | 85.7 | 70.2 | 79.3 | 81.3 |
| FISH Mask | 0.50% | 93.1 | 94.7 | 86.5 | 85.9 | 61.6 | 87.1 | 86.5 | 71.2 | 80.2 | 82.6 |

Table 1: GLUE test server evaluation results with BERT$_{LARGE}$. MRPC and QQP are reported as an average of F1-score and accuracy, and STS-B is reported as an average of Pearson and Spearman correlation. Accuracy is reported for all other tasks. All results are reproduced experimentally. Training with the FISH surpasses (82.6) other methods and equals dense fine-tuning performance (82.5) whilst updating only 0.5% of model parameters.

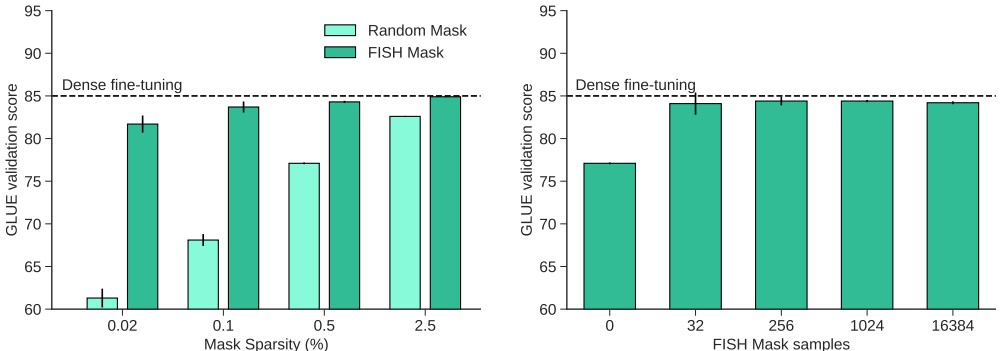

Figure 2: (Left) GLUE validation performance of a randomly selected mask and the FISH Mask at varying levels of mask sparsity. Compared to the densely fine-tuned baseline score of 85%, training with the FISH Mask is competitive at 0.5% mask sparsity. (Right) GLUE validation performance at varying levels of dataset samples used to compute the FISH Mask. Few samples are needed to effectively compute the FISH Mask and obtain good performance. Results in both (Left) and (Right) are averaged over 5 seeds.

the Fisher information in selecting parameters to update. Finally, FISH Mask training is competitive with all other parameter-efficient transfer learning approaches; the next-best score of $81.5\%$ is achieved by Diff Pruning [16]. As mentioned in section 3, we do not include a direct comparison to Adapter-based methods since they add parameters to the model, though we note that our method is able to match BERT$_{LARGE}$'s performance using a significantly lower mask sparsity level (0.5% vs. 3.6%) than the method proposed by Houlsby et al. [19].

Figure 2 shows the change in performance on the GLUE validation set as we change the mask sparsity level for both a random mask and a FISH Mask. We find that the FISH Mask consistently outperforms the random mask baseline and is still strong even at a lower mask sparsity level of $0.1\%$. These results demonstrate that the FISH Mask can be a useful tool in mitigating the storage costs of saving many fine-tuned models since it only requires that the updated parameters and their respective indices are saved.

## 4.2 Distributed Training

Now, we turn to using FISH Mask to reduce communication costs in distributed training settings. We consider the setting where distributed workers compute many subsequent updates to a local copy of the model before transmitting their changes back to a central server. In our experiments, we assume all workers sample data i.i.d. from the same dataset and that all workers compute the same number of updates between each communication step, though our results could carry over to settings where workers use different datasets and varying amounts of updates (e.g. in Federated Learning [35]).

Let $\theta$ denote the parameter vector stored on the central server and $\delta_i$ represent the update computed by worker $i$. After each update/communication step, the server must transmit the updated $\theta$ back to all workers. When standard gradient-based training is used, all parameters are updated, so $|\delta_i| = |\theta|$. If there are $M$ workers, the communication costs for normal training are then $M|\delta_i| = M|\theta|$ for worker-

to-server communication and $M|\theta|$ for server-to-worker communication, for a total communication cost of $2M|\theta|$.

Using a sparse mask will effectively reduce the size of each $\delta_i$ to $k$. Without loss of generality, we assume that $|\delta_i|$ is the same across all workers. Furthermore, assuming updates are aggregated on the central server by summing them together, the server can either communicate the full updated parameter vector back to all workers, or the server can communicate all *other* workers' updates to a given worker. These two options amount to server-to-worker communication costs of either $M|\theta|$ or $M(M-1)\|\delta_i\| \approx M^2|\delta_i|$. In general, we expect $M$ to be small and $|\theta|$ to be large, so we typically have $M^2|\delta_i| < M|\theta|$. Therefore, we can achieve significant communication savings by using sparse updates. For simplicity, we set $M = 2$ in all of our experiments; this makes the savings in communication equal to the mask sparsity level. To apply FISH in distributed training, the central server computes $\hat{F}_\theta$ and a single FISH Mask which is shared across all workers.

**Baselines**    For distributed training experiments, we compare FISH Mask training to three baselines: First, in **standard training**, we tune all the parameters in a single machine (i.e. standard, non-distributed training). This provides an upper bound on performance for distributed training techniques. Second, in **densely-updated distributed training**, workers use standard gradient descent to compute updates over all parameters, and all of the worker's updates are added together on the centralized server. Third, in **random mask distributed training**, we randomly select a subset of parameters to update for workers. For a fair comparison, we keep the overall training batches the same for all methods, so the training iterations of a worker are half of that of the single-machine baseline.

**Experimental setting**    In preliminary experiments on the GLUE benchmark (described in appendix A), we found that densely-updated distributed training saw no real degradation in performance even for long communication delays. This suggests that stale gradients could be less of an issue in transfer learning settings. We therefore instead focused on from-scratch training of a ResNet-34 on CIFAR-10. We train the model for 100 total epochs, with 50 epochs performed by each of the two workers. Notably, we found that the performance of sparse update methods was poor for from-scratch training unless we performed 5 epochs of standard training as "warmup" before beginning distributed training. Beyond this change, all the models and hyper-parameters follow those mentioned in section 4.1. We searched for the best initial learning rate in {0.4, 0.2, 0.08, 0.04, 0.02}. We measure performance using a varying number of parameter updates between each worker-server communication step. We report performance in terms of the accuracy achieved under a certain communication budget, where the communication cost is measured in terms of the equivalent number of full model parameter updates. For example, a method that updates 10% of the model's parameters and performs 5 communication steps over the course of training has the same cost as a method that communicates all of a model's parameters once (since we must transmit both the updated parameters and their locations; more details are in section 4.3).

**Results**    Results from training a ResNet-34 on CIFAR-10 are shown in fig. 3. As in section 4.1, using a FISH Mask works better than using a random mask for all communication costs. Furthermore, we generally find that using FISH Mask with a sparsity level of 10% attains a better commnication/performance trade-off than densely-updated distributed training. For example, FISH Mask training attains comparable performance to standard training when only communicating two copies of the model, whereas densely-updated training performs significantly worse at this communication amount. Notably, performance was relatively poor when only updating 2% of the model's parameters with FISH Mask, suggesting there is a lower bound under which it is difficult to attain reasonable results. As a whole, our results show significant promise for dramatically reducing computational costs in distributed training settings.

## 4.3 Efficient Checkpointing

Over the course of training a machine learning model, it is common to save intermediate *checkpoint* files that store the model's parameter values. These checkpoints can be useful for restarting training from a given iteration rather than starting from scratch in the event that the training job crashes or is otherwise stopped. They are also commonly used for post-hoc analysis, for example for evaluating a model's performance on new datasets or metrics over the course of training. Since checkpoints store a full copy of the model's parameters, they can take up a significant amount of space on disk.

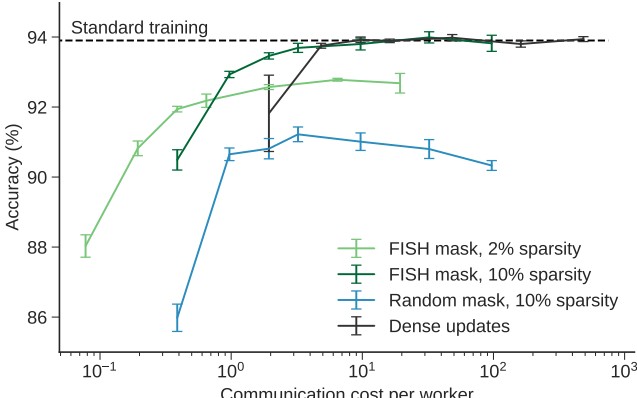

Figure 3: CIFAR-10 validation set accuracy achieved by a ResNet-34 through distributed training at different communication costs. X-axis refers to the total number of model communications required for a single worker. Standard (non-distributed) training achieves an accuracy of 93.9%.

Furthermore, depending on the checkpointing frequency, hundreds of checkpoints are often written to disk over the course of a training run. The development cycle of a machine learning model can result in hundreds of different model variants being trained. Combining these factors with the on-disk space needed to store the parameters of modern models (around 1 GB for BERT$_{\text{LARGE}}$) results in potentially massive storage costs.

Training with sparse updates can significantly reduce these storage costs. Specifically, if only a small subset of the parameters are updated between checkpoint saves, then the checkpoint only needs to store the updated parameter values and indices denoting the position of the updated parameters' values. Assuming that the storage costs for a parameter value and index is the same (e.g. using a 32-bit float for parameter values and a 32-bit integer for indices), using a sparse mask will reduce storage cost when the mask sparsity level is less than 50%. Note that this setting allows the "mask" to change over the course of training, and is therefore a relaxation of the setting in section 4.1, where the requirement is that the same subset of parameters is updated over the entire fine-tuning run. It follows from our results in section 4.1 that FISH Mask training could be readily applied to reducing checkpoint size in parameter-efficient transfer learning. However, we found that the strict requirement of fixing the mask over an entire from-scratch training run on CIFAR-10 resulted in a significant degradation in performance. This is in line with past work demonstrating the difficulty of identifying fixed sparse subnetworks to train before training begins [15]. We therefore focus on from-scratch training on CIFAR-10 and allow the mask to change every time a checkpoint is written, which does not increase storage requirements over using the same fixed mask from the start of training.

Overall, we use the same experimental setup as in section 4.2. We measure performance when updating the mask every epoch (which is a common choice in practice), every 2 epochs, every 4 epochs, and leaving the mask fixed over the course of training. We performed a new search over learning rates in {0.4, 0.2, 0.08, 0.04, 0.02}. We compare to baselines of standard training (which to serve as an upper bound on the performance of using a FISH Mask) and using a random mask.

Table 2: CIFAR-10 validation set accuracy when using the FISH Mask and the Random Mask to reduce checkpoint sizes. "Epoch" refers to allowing the mask to change each epoch, and the number is how many epochs we update masks. Standard training achieves an accuracy of 93.9 ($\pm$0.1)%.

|  | Mask sparsity level | | |
|---|---|---|---|
|  | 0.5% | 2% | 10% |
| Random Mask (1 Epoch) | $74.8_{0.6}$ | $84.4_{0.2}$ | $90.0_{0.2}$ |
| FISH Mask (1 Epoch) | $90.5_{0.3}$ | $93.0_{0.3}$ | $93.9_{0.1}$ |
| FISH Mask (2 Epochs) | $90.3_{0.5}$ | $92.5_{0.1}$ | $93.7_{0.2}$ |
| FISH Mask (4 Epochs) | $89.4_{0.6}$ | $92.1_{0.3}$ | $93.4_{0.2}$ |
| FISH Mask (Fixed) | $78.5_{0.7}$ | $90.6_{0.2}$ | $93.0_{0.2}$ |

**Results** In table 2, we show the results of FISH Mask training when keeping the mask fixed over the course of training or updating it at each epoch. We find that updating the FISH Mask every epoch can match the performance of normal training (93.9% accuracy) at a mask sparsity level of 10%,

which would reduce storage requirements by a factor of 5. At lower mask sparsity levels, we see some degradation in performance. We find that accuracy tends to decrease as we decrease the frequency of updating the mask, but this effect is relatively small. As mentioned earlier, we also found that using a fixed mask significantly degraded performance, though only by a few percent at a mask sparsity level of 10%. This suggests that the FISH Mask could also be useful for identifying sparse subnetworks to train before training begins, as conjectured by the Lottery Ticket Hypothesis [15]. We leave the exploration of this possibility for future work. Lastly, the FISH Mask's performance is unanimously better than the Random Mask across the three mask sparsity levels.

## 4.4   Ablations

Having established the effectiveness of training with a FISH mask, we now ablate a few design choices to help demonstrate the robustness of our approach. We perform all ablation experiments in the parameter-efficient transfer learning setup described in section 4.1.

### 4.4.1   True Fisher vs. Empirical Fisher

In section 2, we note that past work has approximated the Fisher information matrix (eq. (3)) using either the expectation over $y \sim p_\theta(y|x)$ ("true Fisher") or ground-truth labels ("empirical Fisher"). While past work has shown that using the empirical Fisher can be detrimental in optimization settings [34, 26], we mainly use the Fisher information as a signal of parameter importance. The empirical Fisher also has the benefit that it avoids marginalizing over or sampling from $p_\theta(y|x)$ and only requires computing the gradient for a single value of $y$. When comparing the performance of using the true or empirical Fisher to compute a 0.5%-sparse FISH Mask for parameter-efficient transfer learning, we observe that both methods achieve near-identical performance with an average validation-set GLUE score of 82.5 in both cases. Since computing the empirical Fisher can be more computationally efficient, we used the empirical Fisher for all experiments.

### 4.4.2   Sample Ablation

We also ablate the number of samples, $N$, used to compute the FISH Mask to study if more samples are beneficial. The results for parameter-efficient transfer learning on GLUE are shown in fig. 2, right. At a sample count of 0, the FISH Mask is equivalent to the Random Mask baseline presented in section 4.1 in which parameters to update are selected at random instead of informed by the Fisher information. We observe that FISH Mask performance on the GLUE validation set is surprisingly stable across many values of samples, with just 32 samples needed to achieve the highest-possible performance. These suggest that using the approximate Fisher information is a data-efficient approach of computing parameter importance, and we therefore ran all experiments with a sample count $N = 1024$, except in distributed training we use $N = 256$ for efficiency.

## 5   Conclusion

In this work, we proposed FISH Mask training as a novel method for pre-computing fixed sparse masks of a model's parameters to update over many subsequent iterations. The FISH Mask estimates the importance of each of a model's parameters by first approximating the Fisher information of each parameter and then selecting the $k$ parameters with the largest Fisher information to include in the mask. We demonstrate the usefulness of FISH Mask training in several settings, including parameter-efficient transfer learning, distributed training, and reducing storage requirements of model checkpoints. In future work, we hope to explore methods for improving the performance of FISH Mask training at lower mask sparsity levels, possibly by considering other measures of parameter importance. We also hope to further demonstrate the efficacy of FISH Mask in real-world settings where the benefits of sparse parameter updating are even more pronounced, such as in Federated Learning. The integration of FISH Masks across tasks and sharing amongst practitioners could also be a useful line of inquiry, as recent frameworks such as AdapterHub [39] have enabled for Adapter modules [19].

## Acknowledgments and Disclosure of Funding

We thank Yoon Kim, Michael Matena, and Demi Guo for helpful discussions.

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
