# OpenReview forum: "Training Neural Networks with Fixed Sparse Masks"
_NeurIPS.cc/2021/Conference — NeurIPS 2021 Poster_

### Official Review · Reviewer_Z6Qg · 2021-07-14

**Rating:** 6
**Confidence:** 4

**Summary:**

This paper attempts to discover a fixed sparse subnetwork before the standard training. Only training the masked subnetwork can sufficiently match or even exceed the performance of the dense counterpart. The performance achieved by the proposed method is promising.

**Limitations And Societal Impact:**

There are some limitations in the paper that needs to be addressed in the rebuttal. I would consider increasing my score if they are addresses.

**Main Review:**

Originality: The novelty of this paper is limited.

(1) The pruning technique that is used in this paper, Fisher approximations, is already widely used in the pruning literature [1][2]. It is not surprising to see that fisher pruning performs well before training. On the other hand, there is recently an increased number of works on pruning before training, e.g., SNIP [3], GrsSP[4], Synflow [5]. This paper is likely to belong to this field. However, none of these papers is cited and compared properly in the submission.  Fairly, directly applying fisher pruning to the pruning before training field is still valuable if it can outperform or match the above-mentioned methods with higher sparsity. I would highly encourage the authors to compare with them and report the results. Either positive or negative results are valuable for the community.

(2) As the proposed methods are doing sparse training (dense-to-sparse training), an important class of sparse training methods is ignored, that is, dynamic sparse training (DST). Instead of keeping the mask fixed,  DST allows the mask to dynamically change during training, improving the expressibility of the model gradually. Thus, a comparison with the SOTA DST methods such as RigL [6], ITOP [7] is suggested.

Quality:

The paper is easy to understand. However, these are some big mistakes that demonstrate that the authors have limited knowledge of the research field of pruning. As far as I know, the common definition of sparsity refers to the ratio of the pruned/masked parameters to the total number of parameters. The usage of sparsity in this paper is exactly the opposite to it, e.g., line 228. Moreover, there are some typos here and there, e.g., line 118; the color in figure 1 is difficult to discriminate for readers.

Although I like the fisher pruning idea, the carefulness in the current version degrades the quality of the paper significantly.


Reference
[1] Theis, Lucas, et al. "Faster gaze prediction with dense networks and fisher pruning." arXiv preprint arXiv:1801.05787 (2018).
[2] Liu, Liyang, et al. "Group Fisher Pruning for Practical Network Compression." International Conference on Machine Learning. PMLR, 2021.
[3] Lee, Namhoon, Thalaiyasingam Ajanthan, and Philip HS Torr. "Snip: Single-shot network pruning based on connection sensitivity." arXiv preprint arXiv:1810.02340 (2018).
[4] Wang, Chaoqi, Guodong Zhang, and Roger Grosse. "Picking winning tickets before training by preserving gradient flow." arXiv preprint arXiv:2002.07376 (2020).
[5] Tanaka, Hidenori, et al. "Pruning neural networks without any data by iteratively conserving synaptic flow." arXiv preprint arXiv:2006.05467 (2020).
[6] Evci, Utku, et al. "Rigging the lottery: Making all tickets winners." International Conference on Machine Learning. PMLR, 2020.
[7] Liu, Shiwei, et al. "Do we actually need dense over-parameterization? in-time over-parameterization in sparse training." International Conference on Machine Learning. PMLR, 2021.

**Time Spent Reviewing:**

6 hours

---

> ### Author Response · Authors · 2021-08-10
> **Response to Reviewer Z6Qg**
>
> Thank you for the detailed review. The following are our replies to your comments and concerns.
>
> * ***"(1) The pruning technique that is used in this paper, Fisher approximations, is already widely used in the pruning literature [1][2]... (2) As the proposed methods are doing sparse training (dense-to-sparse training), an important class of sparse training methods is ignored,..."***
>
> Like some of the pruning papers you referenced (e.g. [1, 2]), we use the Fisher Information to identify a sparse subset of important parameters. However, unlike work on network pruning or sparsification, the focus of our paper is not pruning or training neural networks with sparse weights. Instead, it focuses on selecting a fixed, sparse subset of weights in a non-sparse network to update over the course of training. Our application domain includes the problems of parameter-efficient transfer learning and communication-efficient distributed training.
>
> We agree that a better discussion of the related field of network pruning would be helpful, as well as a specific discussion of how our application setting is different from pruning and network sparsification. We will add an additional paragraph to our related work section on this class of techniques.
>
> * ***"However, these are some big mistakes that demonstrate that the authors have limited knowledge of the research field of pruning...sparsity refers to the ratio of the pruned/masked parameters to the total number of parameters…"***
>
> Thanks for pointing out the confusion of our use of the term “sparsity.” As per our clarification above, the focus of the research is different than network pruning, so some terms may not follow the convention of that field. Instead, our definition of “sparsity” in this paper follows a closely related work ([3]),  in which “sparsity” defines the percentage of non-zero parameters selected for training by the FISH mask. However, to avoid confusion, we will change to use “mask sparsity” rather than “sparsity”.
>
> **Comment References**
>
> [1] Theis, Lucas, et al. "Faster gaze prediction with dense networks and fisher pruning." arXiv preprint arXiv:1801.05787 (2018).
>
> [2] Liu, Liyang, et al. "Group Fisher Pruning for Practical Network Compression." International Conference on Machine Learning. PMLR, 2021.
>
> [3] Guo, Demi et al. “Parameter-Efficient Transfer Learning with Diff Pruning.” ACL/IJCNLP (2021).

---

> > ### Comment · Reviewer_Z6Qg · 2021-08-23
> > **Thank you for your response**
> >
> > I missed the part of the non-sparse network during fine-tuning. This indeed differs from the goal of sparse training. Thus, I raised my score to 6: Marginally above the acceptance threshold, due to the simple but effective idea used.
> >
> > Moreover, I am looking forward to seeing the difference between your method and sparse training/pruning in the next version.

---

### Official Review · Reviewer_citk · 2021-07-15

**Rating:** 7
**Confidence:** 2

**Summary:**

In this paper, the authors propose a scheme to select and update only part of the weights of a neural network during training. The selection of the weights to update is done using an approximation of the Fisher information, and the mask is computed by keeping only the weights associated with the larger fisher information values. The authors then propose an extensive empirical evaluation of their method for transfer learning, distributed training and efficient checkpointing. Finally, the authors perform an ablation study to show the effect of their design choices.

**Limitations And Societal Impact:**


- **L1:** One major limitation I see for the proposed method is that it does not seem to apply to regression. The full paper seems to have been written with classification task in mind, where the logits are amendable to the score function and all experiments are performed for classification tasks. Could the authors comments on the applicability of their method to the regression setting? Indeed, in this case, the score function $p(y | x$) is not directly accessible. If the method cannot be applied in this setting, this limitation should be made clear in the paper.

**Main Review:**

Review Summary
--------------

As a disclaimer, I am not an expert for neural networks training and in particular, I might have missed part of the newest literature.
The contribution of the paper, while not very surprising, is well explained and evaluated. Some improvement could be made with the limitation (**L`1**) and the discussion of the results (**M1**) but I feel that this paper is worth accepting to NeurIPS.

Major comments and questions
----------------------------

The paper is clear and the main idea is well explained. The numerical experiments seems convincing, in particular for the transfer learning ones. I have a few comments and clarifications I would like to ask:

- **M1:** In the results comments for distributed learning and efficient checkpointing, while the performance drop by 0.5 to 1%, the authors state that the performance are comparable. To make this statement clearer, I think repeating the experiment and adding standard deviation would improve the results. Else, I would state that there is a small performance loss here.

- **M2:** I feel like this experiments are a bit unfair for the proposed method, in particular with distributed training and small checkpoint experiments. Indeed, the idea of training only part of the weights for transfer learning makes sense but for a randomly initialized network, training for a few tens of epochs before switching to training only the most important weights (as decided with the FISH mask) would make sens. It seems that the authors implemented such feature with the `warmup_epochs` parameter but it is unused in the script they provided. Could the authors comments on this?

- **M3:** A nice addition to Figure.2 (left) would be to add the performance of the random mask for different sparsity levels too. This way, one would see the value of selecting the coordinates with the proposed methodology.


Minor comments, nitpicks and typos
----------------------------------

- l.118: Broken hyperlink.

- References: It seems that this line of work is somewhat linked to block coordinate descent algorithms and active set ideas in classical convex optimization. The core idea in this two line of work is to only update part of the parameters at once. In particular, the idea of using the Hessian information -- which is linked to the fisher matrix -- to select and split the coordinate between workers in distributed optimization has been proposed in the context of sparse optimization [A]. I feel that highlighting the connection between the two fields would probably be beneficial to draw attention to this line of work from both side.

Extra references
----------------

[A]: Scherrer, C., Tewari, A., Halappanavar, M. & Haglin, D. J. [Feature Clustering for Accelerating Parallel Coordinate Descent](https://proceedings.neurips.cc/paper/2012/hash/4e732ced3463d06de0ca9a15b6153677-Abstract.html). in Advances in Neural Information Processing Systems (NeurIPS) 28–36 (2012).

**Time Spent Reviewing:**

6hours

---

> ### Author Response · Authors · 2021-08-10
> **Response to Reviewer citk**
>
> Thank you for the comprehensive comments. Our replies to your concerns are shown below.
>
> * ***"M1:  In the results comments for distributed learning and efficient checkpointing, while the performance drop by 0.5 to 1%,..."***
>
> We will report standard deviations over 5 runs. We also ran experiments on a larger ResNet34 after submission and found that the performance gap was even smaller, closer to 0.3%. We hypothesize that the performance drop in ResNet9 is due to the model’s significantly smaller size; we leave investigation of this factor to future work.
>
> * ***"M2: I feel like this experiments are a bit unfair for the proposed method, in particular with distributed training and small checkpoint experiments…"***
>
> In CIFAR10, when the FISH mask is fixed, a “warm-up period” of dense updates helped to increase the accuracy by around 1%. However, this result is still lower than when the mask was allowed to update. On the other hand, we did not observe a major improvement from using a “warm-up period” of dense updates when the mask was allowed to change over the course of training (i.e. in the distributed training and efficient checkpointing experiments). So, we did not use this option in any experiments.
>
> * ***"M3: A nice addition to Figure.2 (left) would be to add the performance of the random mask…."***
>
> We are running all random baselines, and we agree that comparing the FISH masks to random masks with different mask sparsity will be helpful to demonstrate the usefulness of the FISH mask.
>
> * ***"References: It seems that this line of work is somewhat linked to block coordinate descent algorithms and active set ideas in classical convex optimization…."***
>
> Thanks for providing the insights to connect our work to that research. We will add related references to this paper.
>
> * ***"L1: One major limitation I see for the proposed method is that it does not seem to apply to regression…."***
>
> The FISH mask is applicable to any setting where the classifier is outputting a probability distribution over a label variable. For regression tasks, we use the standard interpretation of a Gaussian-distributed output variable, that is \\( y=f(x)+\epsilon \\), where \\( f(x) \\) is the model's output and \\( \epsilon \sim N(0, \sigma) \\). We have included results from experiments on STS-B, a regression task, in the parameter-efficient transfer learning setting (see Table 1).

---

> > ### Comment · Reviewer_citk · 2021-08-26
> > **Post Rebuttal comments**
> >
> > I thank the authors for their feedback and clarification.
> > I think the proposed modifications to the figure will improve the quality of the manuscript and I keep my recommendation to accept.
> >
> > As for **L1**, I thank the authors for the clarification. Adding a small remark saying that all task in GLUE are single sentence or sentence pair classification, except STS-B, which is a regression task would probably help the reader that is not familiar with this benchmark. Also, adding a comment on this point at the end of `Section 2.1` would be nice as taking noise models that are non Gaussian might be complicated in this part.

---

### Official Review · Reviewer_wqY5 · 2021-07-16

**Rating:** 6
**Confidence:** 3

**Summary:**

This work introduces FISH - a method of training (or fine-tining) a neural network by updating only a small fraction of its parameters. FISH uses an approximation of the Fisher Information of each of the network’s parameters to determine the top $k$ ones to update. The method is evaluated via transfer learning and distributed learning tasks using GLUE and CIFAR-10 datasets. The experimental results show that FISH has competitive or better performance in comparison to existing methods by updating just a small fraction of a networks’ parameters at a time. Further, FISH can reduce the memory usage when saving checkpoints.

**Limitations And Societal Impact:**

Not explicitly. The limitations could include computational overhead when computing the Fisher information for some particular datasets for which its approximation is not sufficient. Potential negative impact on society can occur if the underlying model that is trained using FISH is biased or has unethical applications.

**Main Review:**

### Strengths
- The idea of using an approximation to the Fisher Information of each parameter for selecting the parameters that have the largest impact on the output is well motivated.
- The proposed model with FISH mask outperforms other benchmarks while having a lower level of sparsity, including a random mask (and with the exception of Bit-Fit in Table 1 which is harder to compare to since its sparsity is much lower).
- To the best of my knowledge, the paper provides a comprehensive literature review and background.
The paper is organized and written clearly.

### Room for improvement and questions
1. What is the time/complexity of computing the (approximate) Fisher Information? The text mentions that it is computationally efficient but does not elaborate further.
2. The work would benefit from reporting error bars/running experiments with several random seeds.
3. Have the authors explored how the frequency of mask computation affects the models’ performance (other than using a fixed mask throughout training and updating it at every epoch)?
4. Are the FISH models in Fig 2 trained with a fixed mask or with a mask which is updated at every epoch?
5. Is it feasible to use the FISH mask in the case of unsupervised learning?

### Nitpicks
- Line 92: “over N samples”
- Line 94: $F_{\theta}$ should be $\hat{F}_{\theta}$
- Line 118: missing reference
- Line 177: “of which” should be removed
- Line 213: “server. Results”



**Time Spent Reviewing:**

5

---

> ### Author Response · Authors · 2021-08-10
> **Response to Reviewer wqY5**
>
> Thank you for your thorough review and comments! The following are our replies to your comments and questions:
>
> * ***“..and with the exception of Bit-Fit in Table 1 which is harder to compare to since its sparsity is much lower”***
>
> Thank you for pointing this out - we will re-run our algorithm at a sparsity level of 0.08% and add test results to Table 1 for a more direct comparison. In the meantime, we can report the following result from our pre-existing experiments: The GLUE validation set average scores for FISH Mask at 0.10% sparsity and for BitFit are 84.4 and 83.6 respectively. As a result, we are confident that our method will outperform BitFit even at this lower mask sparsity level.
>
>
> * ***"What is the time/complexity of computing the (approximate) Fisher Information? The text mentions that it is computationally efficient but does not elaborate further."***
>
> Computing the FISH Mask over N samples has roughly the same cost as training on N examples, since the mask computation involves calculating the squared gradients (the “empirical Fisher”, see section 2.1) across N samples. In Figure 2, we show that good performance can be attained by computing the FISH Mask across just 32 samples, and all other experiments with the FISH Mask use a default of 1024 samples. This is dramatically fewer examples than is typically seen during the course of normal training; thus, this cost is negligible. Additionally,  the training cost once the FISH Mask is computed is identical to that of standard dense fine-tuning, and is 1.5-2x more memory-efficient than a closely related work (Diff-Pruning [1], see second bullet linked [here](https://openreview.net/forum?id=E4PK0rg2eP&noteId=dU6juEdvzA) for note of memory usage) in the parameter-efficient fine-tuning setting.
>
>
> * ***"The work would benefit from reporting error bars/running experiments with several random seeds."***
>
> We agree that we would like to report error bars, especially in Table 1, however, the GLUE benchmark test server disallows reporting results on multiple runs. We will report error bars and average performance across multiple seeds for Figure 2, which reports GLUE validation scores, and are currently running these experiments.
>
>
> * ***"Have the authors explored how the frequency of mask computation affects the models’ performance (other than using a fixed mask throughout training and updating it at every epoch)?"***
>
> Our distributed training experiments show the impact of computing the mask less often, and we do see a performance decrease as we update the mask less often (see Table 2 and Table 3, in which a larger interval between updates results in poorer performance). In our checkpointing experiments, we choose to update the mask after each epoch since this is the typical interval for saving a checkpoint. To investigate this further, we will also report results when updating the FISH Mask every 2 and 4 epochs.
>
>
> * ***"Are the FISH models in Fig 2 trained with a fixed mask or with a mask which is updated at every epoch?"***
>
> These models are trained with a fixed mask.
>
> * ***"Is it feasible to use the FISH mask in the case of unsupervised learning?"***
>
> It would be feasible. For example, in self-supervised learning, the Fisher approximation could be informed by the proxy labels used in self-supervised tasks. The FISH Mask could also be used in settings where labels are not available by using the “true” Fisher, which approximates the expectation over labels used in the Fisher calculation by sampling from the model (see section 2.1 for more discussion of this method).
>
>
> We also appreciate the comments on the notation and any typos and will correct those appropriately.
>
> **Comment References**
>
> [1] Guo, Demi et al. “Parameter-Efficient Transfer Learning with Diff Pruning.” ACL/IJCNLP (2021).

---

> > ### Comment · Reviewer_wqY5 · 2021-08-23
> > **Thank you!**
> >
> > Thank you for your detailed response and clarifications! The updates to the manuscript you are planning to make sound good. I suggest it would also be valuable to add a comment on the model's applicability to self-supervised learning as well as a comment on any potential negative societal impacts of your work. I look forward to reading the final version!

---

### Official Review · Reviewer_bsUF · 2021-07-24

**Rating:** 6
**Confidence:** 4

**Summary:**

The paper describes a method based on Fisher-information to estimate the importance of different network parameters. Once the top-k parameters in terms of importance are identified, the authors only update these parameters during training. The authors demonstrated the usefulness of their approach in transfer learning experiments (fine-tune only the task-important parameters), distributed/federated learning experiments (to reduce the communication volume through communicating only the sparse updates), and model checkpointing (by updating only a subset of parameters each epoch, and storing the deltas instead of the full parameter tensors at each epoch).

**Limitations And Societal Impact:**

Yes

**Main Review:**

The paper is clearly written. I find the idea simple and practically relevant. The experimental results are convincing. Novelty is rather limited, however, as estimating parameter importance using Fisher information (the squared norm of the parameter gradient) has been explored before (see [1] for example).

Minor comments:
- L118 : Missing reference
- For clarity, I recommend the author use the term ‘mask sparsity’ instead of ‘sparsity’. The latter term is typically used to refer to weight sparsity (i.e, the percentage of weights that are zero), and refers to a body of literature (sparse training, pruning, etc..) that is different from this work.

Overall, I think the paper is borderline as it does not offer any new insights or novel methodologies. However, the practical application of the simple idea presented (pre-compute an update mask based on Fisher information to only update the important parameters) could prove generally useful.

[1]Kirkpatrick, James, et al. "Overcoming catastrophic forgetting in neural networks." Proceedings of the national academy of sciences 114.13 (2017): 3521-3526.

**Time Spent Reviewing:**

2

---

> ### Author Response · Authors · 2021-08-10
> **Response to Reviewer bsUF**
>
> Thank you for your time and comments! Our responses to your points are below:
>
> * ***"Novelty is rather limited, however, as estimating parameter importance using Fisher information (the squared norm of the parameter gradient) has been explored before (see [1] for example)"***
>
> Fisher Information has indeed frequently been used as a measure of parameter importance ([1, 2, 3]). Kirkpatrick et al., 2016 [1]  introduce elastic weight consolidation (EWC), an algorithm that makes use of the Fisher Information matrix to overcome catastrophic forgetting. While we also use the Fisher information to estimate parameter importance, our algorithm is quite different from EWC (we threshold the Fisher information to obtain a sparse mask of parameters to update rather than use the Fisher information to weight a L2-distance based regularizer) and our target applications are different too (parameter-efficient transfer learning and communication-efficient optimization, rather than continual learning). We believe our proposed algorithm and target applications sufficiently differ from past work, while sharing the motivation and benefit from the use of the Fisher.
>
>
> * ***"L118 : Missing reference"***
>
> We have noted this and will be sure to add this in the next version.
>
>
> * ***"For clarity, I recommend the author use the term ‘mask sparsity’ instead of ‘sparsity’. The latter term is typically used to refer to weight sparsity (i.e, the percentage of weights that are zero), and refers to a body of literature (sparse training, pruning, etc..) that is different from this work."***
>
> Thank you for pointing this out, this was also mentioned to us by another reviewer. Our definition of “sparsity” in this paper follows a closely related work ([4]), in which “sparsity” defines the percentage of non-zero parameters selected for training by the FISH mask. However, to avoid confusion, we will change our usage in the paper to use “mask sparsity” rather than “sparsity” as you suggest.
>
>
> **Response References**
>
> [1]Kirkpatrick, James, et al. "Overcoming catastrophic forgetting in neural networks." Proceedings of the national academy of sciences 114.13 (2017): 3521-3526.
>
> [2] M. Tu, et al., "Ranking the parameters of deep neural networks using the fisher information," 2016 IEEE International Conference on Acoustics, Speech and Signal Processing (ICASSP), 2016, pp. 2647-2651, doi: 10.1109/ICASSP.2016.7472157.
>
> [3] Tu, Ming et al. “Reducing the Model Order of Deep Neural Networks Using Information Theory.” 2016 IEEE Computer Society Annual Symposium on VLSI (ISVLSI) (2016): 93-98.
>
> [4] Guo, Demi et al., “Parameter-Efficient Transfer Learning with Diff Pruning.” ACL/IJCNLP (2021).

---

### Decision · Program_Chairs · 2021-09-27

**Decision:**

Accept (Poster)

**Comment:**

The paper studies an approach to pre-selecting a subset of parameters of a neural network to update during training. The idea is to look at (an empirical approximation to) the Fisher information of each parameter, and choose the parameters with the k-largest Fisher information values. Only these parameters are updated during training. The paper evaluates this idea for transfer learning (updating only a subset of the parameters for new tasks), sparse training from scratch, distributed training, and efficient checkpointing (where we save a sequence of sparse updates to the model). The proposed method achieves performance similar to dense training, with fewer updated parameters (between 0.5% and 2% depending on the task), and some performance improvements compared to several previous sparse mask generation techniques.

The reviewers all appreciated the paper’s simplicity, clarity and reasonably thorough experimental validation. Questions raised during the review included the novelty of the work (given that the Fisher information has been widely used as a measure of parameter importance, including for neural network pruning), and the relationship to pruning / sparsification methods. The authors’ response points out that although the use of Fisher information is not a novelty of this paper, the setting is different: this paper selects a subset of parameters to update, rather than selecting a subset of parameters to retain, which makes it applicable to transfer learning in dense models.

After several rounds of discussion, the reviewers converged to a recommendation to accept. While there are some limitations to the novelty of the submission, the proposed approach is practical, clearly described and well-substantiated.